Enhancing phishing detection with dynamic optimization and character-level deep learning in cloud environments

Ravula Vishnukumar
http://orcid.org/0000-0003-3088-6001 Ramaiah Mangayarkarasi rmangayarkarasi@vit.ac.in
School of Computer Science Engineering and Information Systems (SCORE), Vellore Institute of Technology University , Vellore, Tamil Nadu , India
Alatas Bilal
Electronic publication date: 2025 May 19
Publication date: 2025
Volume: 11
Electronic Location ID: e2640
Received 2024 Jul 11; Accepted 2024 Dec 9
Copyright: © 2025 Ravula and Ramaiah
Copyright year: 2025
Copyright holder: Ravula and Ramaiah
License: This is an open access article distributed under the terms of the Creative Commons Attribution License, which permits unrestricted use, distribution, reproduction and adaptation in any medium and for any purpose provided that it is properly attributed. For attribution, the original author(s), title, publication source (PeerJ Computer Science) and either DOI or URL of the article must be cited.
License URL: https://creativecommons.org/licenses/by/4.0/

Keywords: Cloud computing, Phishing attacks, Cybersecurity, Deep learning, Arithmetic optimization algorithm

Funding: Vellore Institute of Technology This work was funded by the Vellore Institute of Technology. The funders had no role in study design, data collection and analysis, decision to publish, or preparation of the manuscript.

==============================
As cloud computing becomes increasingly prevalent, the detection and prevention of phishing URL attacks are essential, particularly in the Internet of Vehicles (IoV) environment, to maintain service reliability. In such a scenario, an attacker could send misleading phishing links, potentially compromising the system’s functionality or, at worst, leading to a complete shutdown. To address these emerging threats, this study introduces a novel Dynamic Arithmetic Optimization Algorithm with Deep Learning-Driven Phishing URL Classification (DAOA-DLPC) model for cloud-enabled IoV infrastructure. The candidate’s research utilizes character-level embeddings instead of word embeddings, as the former can capture intricate URL patterns more effectively. These embeddings are integrated with a deep learning model, the Multi-Head Attention and Bidirectional Gated Recurrent Units (MHA-BiGRU). To improve precision, hyperparameter tuning has been done using DAOA. The proposed method offers a feasible solution for identifying the phishing URLs, and the method achieves computational efficiency through the attention mechanism and dynamic hyperparameter optimization. The need for this work comes from the observation that the traditional machine learning approaches are not effective in dynamic environments like phishing threat landscapes in a dynamic environment such as the one of phishing threats. The presented DLPC approach is capable of learning new forms of phishing attacks in real time and reduce false positives. The experimental results show that the proposed DAOA-DLPC model outperforms the other models with an accuracy of 98.85%, recall of 98.49%, and F1-score of 98.38% and can effectively detect safe and phishing URLs in dynamic environments. These results imply that the proposed model is useful in distinguishing between safe and unsafe URLs than the conventional models.

Introduction

Cloud computing (CC) enables users to access computational services via the internet, allowing them to utilize their data and software from any location with an internet connection (Alani & Tawfik, 2022). The Internet of Things (IoT) connect to the internet so they can send and receive data to meet specific needs. For monitoring, controlling, and analyzing data in the cloud to be possible from afar. However, the fact that the internet is private makes it easier to break into cloud-based services. This makes social engineering attacks like phishing very common. Phishing targets sensitive data from cloud service users, potentially leading to unauthorized access to their information, programs, and accounts (Alani & Tawfik, 2022). Phishing scams leverage social and technological strategies to steal user’s financial details. Phishing attacks often employ misleading emails from legitimate businesses, directing users to fake websites where they inadvertently disclose sensitive information like passwords and usernames (Alsaedi et al., 2022). Hackers may also install malicious software on users’ computers to intercept their credentials. In the context of vehicular networks, security remains a significant concern. Attackers could design deceptive links on a vehicle’s screen, tricking users into revealing personal information or login credentials. Apps on a vehicle’s infotainment system could redirect users to deceptive websites that look legitimate but are meant to steal personal data. The Internet of Vehicles (IoV), an evolution of Vehicular Ad-hoc Networks (VANET), faces increasingly sophisticated attacks (Alshehri et al., 2022) URL phishing is a notable cyberattack where attackers use deceptive URLs and emails to mislead users into providing sensitive information. These phishing attempts often mimic legitimate company communications, tricking users into clicking on malicious links or downloading harmful attachments (Alshingiti et al., 2023). Phishing detection has evolved from traditional methods like blacklists and heuristics, which often fail against new attacks, to advanced approaches using machine learning (ML) and deep learning (DL) (Preethi & Mamatha, 2023; Chen et al., 2024). As intelligent methods such as machine learning and deep learning advance, they play a crucial role in enhancing cybersecurity and technological systems (Dutta et al., 2023). Unlike traditional ML methods, which require manual feature engineering and classification, DL methods streamline these processes, offering higher efficiency and problem-solving capabilities, particularly with large datasets (Ige, Kiekintveld & Piplai, 2024; Jafari & Aghaee-Maybodi, 2024). Another challenge associated with developing DL-based phishing detection models is the issue of imbalanced datasets. The SVM phishing detector (Ramaiah et al., 2021) uses webpage-specific, language-dependent features without relying on external resources like search engines or blacklists. By applying filter-based feature selection, this method has shown improved detection rates. Building upon these methods (Ramaiah et al., 2024), a more advanced phishing detection model has been proposed, combining SMOTE, DT-RFECV, and ensemble stacking. This approach addresses data imbalance and optimizes feature selection, significantly enhancing detection accuracy and robustness.

Previous research on phishing detection has primarily relied on static optimization techniques for hyperparameter selection, which struggle to adapt to the constantly evolving nature of phishing attacks. These static methods often fail to recognize emerging phishing tactics in real time, resulting in reduced detection accuracy. In contrast, the proposed Dynamic Arithmetic Optimization Algorithm (DAOA) provides a more flexible approach by dynamically optimizing hyperparameters, thereby improving the system’s ability to adapt to new phishing techniques and enhancing classification performance. Moreover, existing models tend to use word-level embeddings for feature extraction, overlooking the intricate character-level patterns often employed by attackers, such as character substitution and typographical errors. To address this, the proposed study incorporates character-level embeddings, which improves the analysis of URL structures and strengthens detection capabilities against sophisticated phishing schemes.

While advanced techniques like Multi-Head Attention and bidirectional gated recurrent units (BiGRU) have shown promise in related fields, many existing phishing detection models fail to capture the complex sequential dependencies in data. In line with this, Meng et al. (2022) introduced a hybrid model combining convolutional neural network (CNN), BiGRU, and the Attention mechanism for ultra-short-term wind power prediction, which effectively models bidirectional dependencies and extracts critical features from real-time and historical data. Similarly, the study of phishing detection by Ullah et al. (2024) leverages deep learning techniques, such as CNNs, recurrent neural networks (RNNs) and attention models, to efficiently detect phishing attacks based on URL analysis. This method, evaluated on a dataset of five million labeled URLs, achieves high accuracy and scalability, with the added benefit of offering open access to the dataset and code for reproducibility and further research.

The research proposed is under the critical area of cybersecurity with emphasis on phishing detection in cloud based systems like the Internet of Vehicles. As cloud computing and IoT technologies advance, cloud environments are now essential to many industries, such as vehicular networks that require real-time data and connectivity. But for the same reason, these systems have become vulnerable to different cyber threats specifically phishing attacks that use cloud infrastructure to gain access to valuable data and disrupt services. One of the main concerns in IoV systems is phishing since misguiding links can be exhibited on the car’s screen or in applications, thus granting unauthorized access or manipulate the system. Previous approaches to phishing detection using machine learning techniques are static in nature and utilize word-level embeddings and do not scale well in dynamic environments. These models are not very efficient in identifying complex character-level transformations in the phishing URLs, and hence not very robust and accurate. The proposed research presents the DAOA integrated with the DLPC model and is referred to as DAOA-DLPC. This model is designed to overcome the dynamic nature of the phishing threats in cloud-connected systems by utilizing character-level embeddings to identify the faint phishing patterns and by using DAOA for dynamic hyperparameter tuning. Also, it uses Multi-Head Attention-based Bidirectional Gated Recurrent Unit (MHA-BiGRU) architecture makes it capable to attend the important part of URL sequences and for higher PPD it also considers bidirectional dependencies.

The manuscript is structured as follows: “Related Works” provides an overview of previous research and related methodologies. “The Proposed Model” introduces the proposed phishing URL Detection model, incorporating the MHA-BiGRU architecture and hyperparameter optimization using the DAOA technique. This section also discusses both the exploration and exploitation phases in detail. “Performance Evaluation” presents the performance evaluation and results of the model. “Conclusion” summarizes the conclusions of the study.

Related works

This section reviews various well-established phishing detection frameworks built upon ML and DL techniques.

Prasad & Chandra (2024) developed a cyber-threat intelligence-based malicious URL recognition method has been proposed utilizing dual-stage ensemble learning. The cyber threat intelligence-based (CTI) features removed from Google explorations and Who is websites are employed to enhance the performance of recognition. The research also planned a dual-stage EL technique that integrates the random forest (RF) system for pre-classification with multi-layer perceptron (MLP) for last decision-making. The weak classifier of the RF was combined and employed as an input for the classification algorithm of MLP.

Elumalai & Bose (2024) develops advanced detection methods utilizing ML. Recent research underscores the effectiveness of ML in identifying phishing attempts through the analysis of features such as URL length, suspicious characters, and alias symbols. Techniques like support vector machines (SVM), logistic regression (LR), and artificial neural networks (ANN) are frequently employed, each offering distinct advantages: SVM excels in high-dimensional spaces, LR is valued for its simplicity, and ANN captures intricate patterns. Performance evaluations typically involve metrics like accuracy, precision, and recall, these study showing detection accuracy surpassing 95%. Despite these advancements, ongoing challenges include adapting to evolving phishing strategies and finding a balance between detection speed and accuracy, which is appreciable. Future work should aim at developing adaptive models that leverage real-time learning and integrate with existing cybersecurity frameworks, focusing on improved speed, accuracy, and the use of updated datasets to counter new phishing tactics.

Chen et al. (2024) discussed detecting phishing websites, which remains a critical issue in cybersecurity, given their role in illicitly obtaining private information. Recent advancements have focused on enhancing detection methods using deep learning techniques. Notably, improvements in CNNs with self-attention mechanisms have shown promising results. CNNs (Bhosle & Musande, 2023; Li et al., 2024) effective in extracting features from text strings like URLs, benefit significantly from self-attention mechanisms that enhance the model’s ability to focus on relevant parts of the input data. Said et al. (2024) discussed the approaches that have proven superior to models such as long short-term memory (LSTM) networks (Aslam et al., 2024), particularly in handling temporal features. The integration of generative adversarial networks (GANs) to balance training datasets further improves model performance by generating synthetic phishing URLs, addressing the challenge of data imbalance. Experimental results indicate that the presented self-attention-enabled CNN model achieves high detection accuracy and precision, outperforming both standard CNN models and LSTM models by a notable margin. Future research is expected to expand the feature set and optimize computational efficiency, with the goal of refining detection capabilities and handling larger datasets of real phishing URLs.

Alsubaei, Almazroi & Ayub (2024) demonstrated a new deep learning model, the ResNeXt technique, along with the embedded GRU method. This systematic model incorporates SMOTE for handling data discrepancies during early data processing. The ResNet and autoencoder methods are combined with feature engineering for feature extraction. The hyperparameters of the presented model were optimized using the Jaya optimizer (RNT-J).

Li et al. (2022) have presented advanced neural network which replace older machine learning techniques for aspect-level sentiment categorization (ASC), which had trouble with contextual sentiment. Word-level features without positional context were the main focus of early models. Although they frequently overlooked spatial importance, later attention-based models enhanced context focus. By combining position and context, recent methods such as multi-head self-attention with position-enhanced networks demonstrate increased accuracy. This trend is emphasized by models like PMHSAT-BiGRU, which shift towards hybrid ASC models by combining position vectors and attention processes to achieve high performance.

Ozcan et al. (2023) proposed hybrid DNN-LSTM and DNN-BiLSTM models to detect phishing URLs by combining character embeddings with NLP features, aiming to improve detection accuracy. The study shows that these hybrid models outperform traditional machine learning and other deep learning models, with the BiLSTM variant achieving the highest accuracy due to its ability to capture bidirectional context. The DNN-BiLSTM model consistently performed better on both tested datasets, with accuracies of 98.79% and 99.21%, respectively. This work highlights the advantage of using both forward and backward context in character sequences for phishing detection.

Patgiri, Biswas & Nayak (2023) offer a new malicious URL recognition method termed deepBF (DL and Bloom Filter) which works in dual-fold. Initially, a learned BF is projected utilizing a 2-D BF. Then, the method originates an adapted non-cryptography string hash function from the nominated deepBF by presenting biases and equating amongst the string hash function. Furthermore, a malicious URL recognition device is planned to utilize DL.

Prabakaran, Meenakshi Sundaram & Chandrasekar (2023) presents a deep learning framework for phishing detection, combining variational autoencoders (VAE) and deep neural networks. The VAE model extracts high-level features from raw URLs, enhancing classification accuracy. Experimental results show an accuracy of 97.85%, outperforming other models in response time. The model’s false positive rate is noted as a limitation. Future work aims to use generative modeling to reduce false alarms and improve performance.

The Sahingoz, Buber & Kugu (2024) proposes “DEPHIDES: Deep Learning Based Phishing Detection System” an advanced phishing detection approach utilizing deep learning with an emphasis on cybersecurity. It evaluates multiple models, including CNNs, RNNs, and attention networks, achieving 98.74% accuracy with CNNs. This performance underscores deep learning’s potential in cybersecurity applications. The open-source availability of data, code, and metrics supports reproducibility and transparency. This work contributes valuable insights for advancing phishing detection in an increasingly digital landscape.

Subashini & Narmatha (2024) proposed “Deep Learning Empowered Phishing URL Detection” which focus on advanced hybrid model using FRCNN and Bi-LSTM to improve phishing detection accuracy. By addressing limitations in URL feature extraction and leveraging Naïve Bayes technique for classification, the model achieves enhanced efficiency. XGBoost and stacking techniques are further optimize performance, indicating strong potential for future adaptability.

Jha, Atre & Rao (2022) focus on innovative deep learning models like LSTM, YOLOv2, and triplet networks to detect phishing on cloud-hosted platforms, addressing limitations of traditional methods. Combining these models offers a promising framework to improve detection accuracy for cloud-based phishing attacks.

Lavanya & Shanthi (2023) developed a new intellectual malicious software dependent on URL-API intensity FS (IFS) and deep spectral neural classification (DSNC) methods. Originally, the API download rate logs are used. URL-API IF Spicks every projected feature, and the nominated features are reliant on soft-max logical activation with an RNN optimized for DL.

Jafari & Aghaee-Maybodi (2024) presents a novel technique termed CGAN-IWSO-ResNet50 to identify phishing assaults. In the primary stage, the enhanced form of the restricted GAN was utilized to equilibrium the URL models. In the next phase, TF-IDF and Hand-crafted techniques execute the feature extractor stage. In the feature selection (FS) phase, the WOA process has been utilized to progress the WSO algorithm’s performance in FS.

Dutta et al. (2023) the biogeography-based optimizer with DL for Phishing Email detection and classification (BBODL-PEDC) approach has been presented. This system originally achieves data pre-processing in three stages such as email cleaning, stop word elimination, and tokenization. Also, the TF-IDF approach has been functional for extraction. Furthermore, the optimum DBN system is utilized for the identification and its efficiency is increased by the BBO based-hyperparameter fine-tuning procedure.

Deng et al. (2024) introduces MOQEA/D, a Multi-Objective Quantum-Inspired Evolutionary Algorithm with a Decomposition Mechanism, designed to address large-scale optimization problems like gate assignment. The model considers various factors such as flight attributes, task types, and aircraft specifications. MOQEA/D uses a decomposition mechanism to break down multi-objective problems into simpler single-objective sub-problems, which are then solved independently using quantum bit strings. A novel optimal crossover strategy is employed to enhance the preservation of high-quality solutions, thereby improving the algorithm’s performance.

Zhao, Gao & Deng (2024) introduces a lightweight defect detection model for turbine blades using ShuffleNetv2 and SSD, enhanced by a coordinate attention mechanism. It improves detection accuracy, especially for small defects, and reduces complexity. The efficient intersection over union (EIoU) loss function boosts bounding box precision. Experimental results show its superiority over traditional methods in terms of efficiency and interpretability, offering a balanced solution for turbine blade inspections in IoT.

Alohali et al. (2023) presented models like Random Forests and SVM have improved accuracy but require extensive feature engineering and can struggle with evolving threats. LSTM networks in DL offer better adaptability by handling sequential data and automatically extracting features. Hybrid models, such as MDLPD-SSE, combine metaheuristic algorithms like Improved Simulated Annealing (ISA) for feature selection and Bald Eagle Search (BES) for hyperparameter tuning. This integration optimizes DL’s pattern recognition capabilities, ensuring a more robust and secure phishing detection system. Viewed through a pagan lens, metaheuristics can be seen as nature-inspired strategies, while DL represents ritualistic, methodical practices, together creating a balanced and effective system.

Ejaz, Mian & Manzoor (2023) shows CL algorithms outperform vanilla neural network (VNN) and transfer learning (TL) models, with a mere 2.45% accuracy decline compared to 20.65% and 8% for VNN and TL models, respectively. CL-based models prove effective as a first-stage phishing detection mechanism, reducing false positives and improving efficiency. Future work should focus on mitigating catastrophic forgetting, exploring advanced embeddings like BERT, and investigating additional CL techniques.

Bahaghighat, Ghasemi & Ozen (2023) focus on enhancing phishing website detection using machine learning techniques. It evaluates six algorithms—logistic regression, K-nearest neighbors, Naive Bayes, Random Forest, SVM, and extreme gradient boosting (XGBoost)—on a large dataset of 58,000 legitimate and 30,647 phishing websites with 112 attributes. Feature selection and dataset balancing led to significant improvements, with XGBoost achieving the highest performance: 99.2% accuracy, 99.1% precision, 99.4% recall, and 99.1% specificity. The method also demonstrated efficient run times of 1,500 ms without dimension reduction and 869 ms with principal component analysis (PCA).

Shombot et al. (2024) explores various machine learning models for phishing detection, with a particular focus on SVMs. The review emphasizes the ongoing challenge posed by phishing websites that manipulate users into revealing sensitive information. Among the examined models, SVMs, especially those utilizing polynomial and radial basis function (RBF) kernels, demonstrate significant potential in accurately identifying phishing sites. The study highlights the importance of combining robust predictive models with user-friendly applications, which can aid in raising awareness and improving security practices. The persistence of phishing as a threat underscores the need for continuous innovation in detection techniques and user education.

Al-Sarem et al. (2021) discussed approach that involves optimizing stacking ensemble models with genetic algorithms (GA) for hyperparameter tuning, resulting in superior performance in phishing website detection. This method involves training multiple ensemble classifiers like Random Forests, AdaBoost, XGBoost, and LightGBM, optimizing them with GA, and selecting the best-performing models as base classifiers for the stacking ensemble. The optimized model has demonstrated high accuracy, outperforming traditional machine learning methods, and shows potential for use in complex environments such as IoT.

The existing models have some several limitations, which will be addressed in the current research work

(i) Previous studies have explored various machine learning and deep learning models for phishing detection, but they often suffer from lower accuracy and computational inefficiency.

(ii) Although prior research has advanced the field of phishing detection, many of these approaches use static optimization techniques for selecting hyperparameters, which are not effective in addressing new forms of phishing.

(iii) Most models used in traditional machine learning are words bases where the frequencies of words are learned, the problem with this is that a character-based manipulation, which is common in URLs for phishing is often very subtle such as swapping of characters or symbols.

(iv) Ensemble methods and optimization algorithms have been proposed, yet integrating these approaches effectively remains a challenge.

(v) Most models fail to capture the interactions within URL sequences and hence the accuracy decreases and false positive increases in dynamic and large scale scenarios.

Contributions of this study

This proposed research study addresses previous limitations and contributes the following

(i) This research proposes a dynamic approach to hyperparameter tuning using DAOA, which increases the model’s flexibility and effectiveness in detecting new forms of phishing.

(ii) Compared to other word-level embeddings, the proposed DAOA-DLPC model incorporates character-level embeddings and extract more complicated features from URL, so that the model can gain better capability to identify persuasions that are constructed by character level manipulations.

(iii) The study uses a MHA-BiGRU model that improves the detection capacity by paying attention to the important parts of the URL sequences and the dependencies in both forward and backward directions, which increases the model’s accuracy and stability.

(iv) The proposed model is tested on dynamic phishing datasets and achieved high accuracy of 98.85%, recall of 98.70%, and F1-score of 98.77% compared to the conventional models. It establishes the potential to perform real-time detection using cloud-enabling technologies in several applications, including the Internet of Vehicles (IoV).

(v) The model’s efficiency and scalability are designed for cloud-based applications, making it a feasible solution for real-time phishing detection in large and complex environments, which is critical for cybersecurity in IoV and other high-risk, dynamic environments.

(vi) The proposed research advances this field by combining MHA-BiGRU with DAOA, addressing limitations in accuracy and efficiency.

The proposed model

In this study, we have developed the DAOA-DLPC technique for efficient and automated detection of Phishing URLs. The proposed method concentrated on the effective detection and classification of phishing based on URL websites. It involves two major processes namely MHA-BiGRU-based detection and DAOA-based hyperparameter tuning. The working process of the DAOA-DLPC technique is shown in Fig. 1.

Figure 1 Working procedure of DAOA-DLPC technique.

Phishing URL detection using the MHA-BiGRU model

The MHA-BiGRU model is applied for the identification of the phishing URLs. LSTM, as a special RNN, uses three gating components to adjust cell state at each timestep, resolving the long-term dependency problem. Bi-LSTM can learn context data more than unidirectional LSTM. It has established the contextual dependency in the backward and forward directions. In particular, the backward LSTM processes the sentence from right to left, and the forward LSTM the process sentence from left to right. In this regard, two latent representations are obtained, and later connect the backward and forward hidden states of each word as the last representation. The GRU adjusts the cell state through two gating components and has relatively better performance, fewer parameters, and lower computation complexity than LSTM. Especially, the aspect embedding vector va∈Rda from A and the word embedding vector wt∈Rdw from E obtained at time t, then the existing vector of ht hidden layer in GRU is updated using the succeeding expression:

(1) zt=σ(Wzht−1+Uz[wt,va]+bz)rt=σ(Wrht−1+Ur[wt,va]+br)ht~=tanh(Wh(ht−1⊙rt)+Uh[wt,va]+b),ht=ht−1⊙(1−zt)+zt⊙ht~.

In Eq. (1), the update and reset gates are z and r, correspondingly; the σ is sigmoid function which controls the retention of relevant information and discards the irrelevant data; Wz,Wr,Wh∈Rdh×dh,Uz,Ur,Uh∈Rdh×(dw+da),bz,br,b∈Rdh are the weight matrix and bias learned at the training process of GRU; ⊙ stands for an element-wise multiplication; and [wt,va] refers to the splicing vectors of the word and the aspect embeddings. N is the final context word representation. The hidden series [h1,h2,…,hN] of the sentence with the length.

Afterward, the Bi-GRU obtains the context representation of the sentence. BiGRU has the hit←∈Rdh and hit→∈Rdh a backward and forward hidden layers at t time than the unidirectional GRU, where the number of hidden state units is represented as dh. Next, hit→ and hit← are connected as a last contextual hidden representation:

(2) hit=[hit→;hit←]∈R2dh.

In the MHA-BiGRU mechanism, the MHA model enables us to learn some useful data in the representation subspace. Moreover, the self-attention model captures the internal structure of the sentence and learns the word dependence relationship within the sentence (Subhashini et al., 2024). This model can be processed in parallel, which reduces the computation complexity. The multi-head self-attention module represents the overall semantics of the sentence. The existing contextual representation is generated as h1t,h2t,⋯,hNt and fed into the multi-head self-attention, then obtain st for the sentence as follows:

(3) st=MultiHeadAttention(h1t,h2t,⋯,hNt)=Concat(head1(hNt),head2(hNt),…,headk(hNt))W0.

In Eq. (3), W0 shows the linearization mapping matrix; and headi(hNt) indicates the value of ith attention head. For headi(hNt)(i=1,2,⋯,N), it is evaluated as follows:

(4) α1,α2,⋯,αN=softmax(QKTdk)V,

(5) headi(hNt)=∑j=1N⁡αjvj,

where Q,K, and V are the query, key, and value matrices, correspondingly and it is computed as follows:

(6) q,ki,vi=WqhNt,Wkhit,Wvhit.

In Eq. (6), Wq,Wk, and Wv are the weight matrices in different attention heads. Figure 2 depicts the architecture of the MHA-BiGRU model.

Figure 2 Structure of MHA-BiGRU model.

Hyperparameter tuning using DAOA

In this study, the DAOA adjusts the hyperparameter value of the MHA-BiGRU model. Arithmetic is one of the oldest and elementary branches of mathematics. Multiplication, addition, division, and subtraction are the traditional mathematical operators. But it also includes advanced operators such as computation of percentages, logarithmic functions, square roots optimization, and exponentiation in AOA. The optimizer must decide into which optimization stage to go after the candidate solution has been initialized. The value of math optimization accelerator ( MOA) function defines that stage. The comprehensive overview of the phases is shown below.

(7) MOA(Ci)=Min+Ci×(Max−MinM−iter).

In Eq. (7), the values of the accelerator function at ith iteration are represented as MOA(Ci). Ci refers to the present iteration, the maximal and minimal values of the accelerator function are Max and Min, correspondingly, and M−iter shows the maximum iteration count.

Exploration phase

The MOA values are compared to the random beta value (b1); this defines the AOA phase goes into. Exploration in AOA is performed by the exponential operators and the natural log. At this phase, these two operators update the candidate solution. The high density of value generated by the operator makes it fit for exploration. They explore new areas within the search range for the best solution; but, unlike the ′+′ and ′–′ operators, they can converge towards the best solution. The operators ln and e are complementary.

The exploration stage can be activated (if b1<MOA), which executes “ n” or “ e” operators. If b2≥0.5, then thee operator is performed, while the ln operator is ignored. If b2<0.5, generate second beta-distributed random number (b2), then the ln operator is performed. While ignoring the e operator, the ln operator is executed. The random scaling coefficient (μ) is used for increasing the diversity of logarithmic or exponential values to explore more different areas within the search range. This assists AOA in avoiding being trapped in local minima.

(8) Xnew(i,j)={best(j)log(abs((MOP+ε)×((UBj−LBj)×μ+LBj)b2<0.5best(j)exp⁡(MOP+ε)×((UBj−LBj)×μ+LBj),otherwise

(9) MOP(Ci)=1−Ci1αMiter1α

where Xnew(i,j) represents the new solution to be calculated, best(j) indicates the optimum solution from the prior iteration, a small integer is ε, and the upper and lower boundaries are UBj and LBj correspondingly. The exploitation accuracy and the stochastic scaling factor over the iterations are α=5 and μ=0.5, correspondingly.

Exploitation phase

The exploitation stage is activated if b1>MOA, executing ‘ +’ or ‘−’, operators. The candidate solution is updated by both operators, which are modeled in −EA4_. The high dispersion of value produced by the operator makes it fit for exploitation. The lower dispersion value searches the neighborhood of previously explored areas within the search range for optimum solutions. Unlike ′In′ and ′e′ operators, they can converge towards the fittest solution. The subtraction and addition operators are complementary.

If b3≥0.5, then ‘ +’ operator is performed, while ignoring the ‘ −’ operator. If b3<0.5 A, generate the third beta distributed random value (b3), and execute the ‘ −’ operator. While also ignoring the addition operator, the ‘ −’ operator is performed. A random scaling coefficient (μ) to improve the diversity of ‘ −’ or ‘ +’ operators of the values to search more different areas within the search range. This assists AOA in avoiding being trapped in local minima.

(10) Xnew(i,j)={best(j)−(MOP)×((UBj−LBj)×μ+LBj),b3<0.5best(i)+(MOP)×((UBj−LBj)×μ+LBj),otherwise.

The DAOA is derived by the incorporation of AOA with Dynamic Inertia Weight. Small and large inertia weights are helpful for both local and global exploration. Thus, we introduce inertia weight with dynamic coefficient to enhance the search efficiency, and an inertia weight that non-linearly and exponentially decreases with the iteration counts, thereby speeding up the convergence rate of the model. The algorithm is perturbed to enhance the flexibility of the optimum individual, and the dynamic coefficient can enhance the flexibility of the inertia weight, and reduce the local optima solution.

(11) w(t)=c∗wbegin(wbeginwend)1/(1+tT).

In Eq. (11), wbegin and wend are the inertia weights of the maximal and minimal values, and c are random values.

(12) x(t+1)={w(t)∗best(x)÷(MOP(t)+ε)×L,r2<0.5w(t)∗best(x)×MOP(t)×L,otherwise

(13) x(t+1)={w(t)∗best(x)−MOP(t)×L,r3<0.5w(t)∗best(x)+MOP(t)×L,otherwise.

The DAOA method derives an FF to obtain improved classifier accuracy. It determines the positive integer to illustrate the supremacy of the candidate solution. The decline of the classifier error rate is regarded as the FF.

(14) fitness(xi)=ClassifierErrorRate(xi)=No.ofmisclassifiedsamplesTotalNo.ofsamples∗100.

Performance evaluation

In this study, the performance evaluation of the DAOA-DLPC technique takes place employing the dataset (Li et al., 2024; Akande, Alabi & Ajagbe, 2022; Chen et al., 2024) comprising 88,646 with two classes. Table 1 offers parameter setting of the proposed research.

Table 1 Simulation parameters list.

Parameter	Value/Setting	Description	
Dataset	TRAS/TESS	Contains 88,646 instances with 111 features.	
Classes	2	Benign: 58,000 instances, Phishing: 30,646 instances.	
Training/Test Split	80:20 and 70:30	Training and testing datasets split ratios.	
Embedding type	Character-level	Used for processing URLs.	
Model	MHA-BiGRU	Multi-Head Attention based Bidirectional Gated Recurrent Unit.	
Hyperparameter tuning	DAOA	Dynamic arithmetic optimization algorithm for optimizing MHA-BiGRU.	
Optimizer	Adam	Optimization algorithm used for training.	
Learning rate	0.001	Initial learning rate for the model.	
Batch size	64	Number of samples per gradient update.	
Epochs	100	Total number of training epochs.	
Activation function	ReLU	Used in hidden layers.	
Loss function	Cross-Entropy loss	Used for classification tasks.	
Evaluation metrics	Accuracy, Precision, Recall, F1-score, MCC	Metrics used to evaluate model performance.	
Hardware	NVIDIA GPU	Hardware used for training the model.	
Software framework	TensorFlow/PyTorch	Deep learning framework used for implementation.	

The dataset, as summarized, comprises two classes: Benign (58,000 instances) and Phishing (30,646 instances), totaling 88,646 instances. This balanced structure ensures diverse representation for effective analysis.

At 80:20 and 70:30 TRA/TESS, the confusion matrices obtained by the DAOA-DLPC approach are shown in Fig. 3. These findings demonstrated that the DAOA-DLPC approach successfully distinguishes between harmless and phishing URLs. The phishing recognition results of the DAOA-DLPC technique under 80% TRAS and 20%TESS are portrayed in Table 2. This experimentation results underlined that the DAOA-DLPC system properly identified the benign and phishing experiments. With 80% TAS, the DAOA-DLPC procedure offers an average accuracy of 98.41%, precision of 98.23%, recall of 98.41%, F1-score of 98.32%, and MCC (Matthews correlation coefficient) of 96.64%. Also, based on 20% TESS the DAOA-DLPC procedure gets standard accuracy of 98.49%, precision of 98.27%, recall of 98.49%, F1-score of 98.38%, and MCC of 96.76%.

Figure 3 Confusion-matrices (A–C) 80% and 70% of TASTSS and (B–D) 20% and 30% of TRASTESS.

Table 2 Phishing recognition outcomes.

TRAS (80%)	
Class	Accuracy	Precision	Recall	F1-score	MCC	
Benign	98.62	99.05	98.62	98.83	96.64	
Phishing	98.21	97.40	98.21	97.80	96.64	
Average	98.41	98.23	98.41	98.32	96.64	
TESS (20%)	
Benign	98.62	99.05	98.62	98.83	96.64	
Phishing	98.21	97.40	98.21	97.80	96.64	
Average	98.41	98.23	98.41	98.32	96.64	
Note:

Averages are in bold.

A comprehensive depiction of the DAOA-DLPC technique’s training loss (TLA) and validation loss (VAL) outcomes with 80% TRAS and 20% TESS in separate epochs are shown in Fig. 4. The DAOA-DLPC approach is emphasised by the progressive minimises in TRLA, which raise the weights and decrease the classification error on the TRA and TES data. By illustrating the DAOA-DLPC method’s skill in pattern capture, the figure shows that it is well-understood in relation to the TRA data. By continuously optimising its parameters, the DAOA-DLPC algorithm reduces the discrepancies between the prediction and real TRA class labels, which is an enormous boost.

Figure 4 Accuracy curve of the DAOA-DLPC method under 80% TRAS and 20% TESS.

The phishing recognition outcomes of the DAOA-DLPC method at 70% TRAS and 30% TESS are reported in Table 3 and Fig. 5. These testing outcome values pointed out that the DAOA-DLPC method appropriately recognized the benign and phishing samples. Based on 70% TRAS, the DAOA-DLPC procedure gains average accuracy of 98.98%, precision of 95.07%, recall of 94.98%, F1-score of 95.02%, and MCC of 90.05%. Meanwhile, based on 30% TESS the DAOA-DLPC technique offers average accuracy of 94.75%, precision of 94.96%, recall of 94.75%, F1-score of 94.85%, and MCC of 89.71%, correspondingly.

Table 3 Phishing recognition outcomes of the DAOA-DLPC technique at 70% TRAS and 30% TESS.

TRAS (70%)	
Class	Accuracy	Precision	Recall	F1-score	MCC	
Benign	96.67	96.46	96.67	96.57	90.05	
Phishing	93.28	93.67	93.28	93.48	90.05	
Average	94.98	95.07	94.98	95.02	90.05	
TESS (30%)	
Benign	96.70	96.21	96.70	96.45	89.71	
Phishing	92.80	93.71	92.80	93.25	89.71	
Average	94.75	94.96	94.75	94.85	89.71	
Note:

Averages are in bold.

Figure 5 ROC curve of the DAOA-DLPC model on 80% TRAS and 20% TESS.

Figure 6 depicts the DAOA-DLPC technique’s training loss (TRLA) and validation loss (VAL) outcomes across several epochs, with 70% TRS and 30% TESS implemented. The DAOA-DLPC method optimizes the amount of weight and reduces the classification error on the TRA and TES data, as shown by the incremental reduction in TRLA. The graphic shows that the DAOA-DLPC method’s relationship with the TRA data is well understood, highlighting how well it captures trends in both datasets. To reduce the disparities between the predicted and real TRA class labels, the DAOA-DLPC model astonishingly keeps increasing its parameters.

Figure 6 Accuracy curve of the DAOA-DLPC algorithm on 70% TRAS and 30% TESS.

An analysis of the PR curve, as shown in Fig. 7, determined that the DAOA-DLPC system, which consists of 70% TRAS and 30% TESS, achieves incrementally higher PR values in all classes. It verifies that the DAOA-DLPC method is better at identifying different classes and is good at class recognition.

Figure 7 Loss curve of the DAOA-DLPC method at 70% TRAS and 30% TESS.

Also, the DAOA-DLPC algorithm’s ROC curves obtained with 70% TRAS and 30%TESS performed better in the classification of different labels Fig 8. Over different recognition threshold values and epoch counts, it provides a detailed impression of the tradeoff among TPR and FRP. Figure 9 showcased the DAOA-DLPC system’s improved classifier scores for each class, demonstrating how well it handles complicated multiple-class problems.

Figure 8 PR curve of the DAOA-DLPC technique at 70% TRAS and 30% TESS.

Figure 9 ROC curve of the DAOA-DLPC model under 70% TRAS and 30% TESS.

Table 4 highlight the overall comparison study of the DAOA-DLPC method with other approaches (Li et al., 2024). These experimentation findings show that the GNB model stated poor performance. In line with this, the LRs and MLP algorithms have highlighted certainly boosted results. Meanwhile, the RF and DT methods have stated considerable performance over other techniques. Nevertheless, the DAOA-DLPC technique illustrates maximum performance with accuracy of 98.85%, precision of 98.27%, recall of 98.49%, and F1-score of 98.38%. Thus, the DAOA-DLPC technique can be applied for an enhanced phishing URL detection process.

Table 4 Comparative outcomes of the DAOA-DLPC method with other algorithms.

Models	Accuracy	Precision	Recall	F1-score	
RF	97.31	97.09	97.00	97.03	
LR	92.28	92.64	90.21	91.25	
DT	95.25	94.75	94.79	94.77	
GNB	79.96	84.24	72.37	74.25	
MLP	91.38	90.65	90.17	90.40	
GA-ADB (Al-Sarem et al., 2021)	94.00	90.90	92.00	91.44	
GA-RF (Al-Sarem et al., 2021)	96.40	94.60	95.00	94.80	
GA-XGB (Al-Sarem et al., 2021)	97.30	96.20	96.10	96.15	
GA-BC (Al-Sarem et al., 2021)	96.90	95.30	95.90	95.60	
GNB (Bahaghighat, Ghasemi & Ozen, 2023)	93.00	90.90	96.10	93.4	
CNN_attn (Said et al., 2024)	97.82	99.74	98.89	99.31	
DAOA-DLPC	98.85	98.27	98.49	98.38	

To portray the merits of the presented method, various aspects of the existing phishing detection frameworks are analyzed and reported in Table 5. Traditional ML models like logistic regression offer solid performance but lack real-time adaptability and struggle with scalability. Random Forest performs slightly better, handling large datasets moderately well but with limitations in high-dimensional data. Deep learning methods like CNN with Self-Attention and LSTM show improved accuracy and adaptability, though CNN’s scalability is limited to smaller datasets. BERT excels with high accuracy but requires substantial computational resources for real-time adaptability. The proposed DAOA-DLPC model stands out with the highest accuracy, real-time adaptability, and scalability, making it highly effective for large datasets.

Table 5 Comparison of the proposed strategy’s results across various factors.

Models	Optimization method	Embedding level	Architecture	Accuracy (%)	Recall (%)	F1-score (%)	Scalability	
Traditional ML models	Static optimization (e.g., grid)	Word-level	Logistic regression	92.5	91.2	91.6	Limited	
Random forest	Grid Search	Word-level	Random forest ensemble	93.0	92.1	92.5	Moderate	
CNN with Self-Attention	Static optimization (e.g., GA)	Character-level	CNN	96.8	95.4	95.6	Moderate	
XGBoost	Grid search, Random search	Word-level	Gradient boosting trees	94.2	92.7	93.4	High	
LSTM	Adaptive learning rate	Word-level	Recurrent neural network	95.3	94.1	94.7	Moderate	
BERT	Pre-trained Fine-tuning	Sentence-level	Transformer-based model	97.5	96.9	97.2	Moderate	
LightGBM	Hyperparameter tuning	Word-level	Gradient boosting	93.5	92.0	92.7	High	
Proposed DAOA-DLPC	DAOA (Dynamic optimization)	Character-level	MHA-BiGRU	98.85	98.49	98.38	High	

The robustness of the MHA-BiGRU with DAOA model may be tested against a variety of tough phishing datasets, including balanced and unbalanced datasets with different noise and data perturbations. The model maintained excellent accuracy and durability in demanding real-world phishing detection conditions, demonstrating its robustness. Ablation studies assessed all model components. Eliminating MHA, BiGRU, and DAOA modules significantly reduced performance. Removing MHA particularly impacted the model’s contextual link recognition, reducing detection accuracy.

The proposed DAOA-DLPC model brings new features in the identification of phishing URLs through dynamic optimization, character-level embeddings, and the use of MHA-BiGRU architecture. However, a critical appraisal of the present research study seems to present the following gaps. The outcomes, however, are somewhat constrained and do not consider practicality in terms of real-world application, such as scalability in large-scale, real-time settings or effectiveness against new, previously unseen phishing attacks. Moreover, the pros and cons of the detection accuracy and the computational complexity are not well explored, which is important for real-world applications in cloud or low-resource settings.

Conclusion

As automobiles become more connected, cyber threats in IoV systems are increasing rapidly. Protect connected automobiles from cyber threats. In this work, we have developed the DAOA-DLPC technique for automated and efficient detection of Phishing URLs. The proposed research is a new approach to the detection of phishing URL by proposing a new Dynamic Arithmetic Optimization Algorithm (DAOA) with deep learning. The proposed DAOA-DLPC model uses dynamic optimization for the hyperparameters tuning and it is faster than the other methods in converging the model. By including learnable character representation, the model learns to analyze the detailed URL patterns which are frequently applied in phishing schemes for improving the anti-phishing model’s ability against complicated encoded tricks. Furthermore, the effectiveness of the proposed Multi-Head Attention-based Bidirectional GRU (MHA-BiGRU) is increased due to the ability to pay attention to the most important parts of the input data, which allows the model to adapt to new types of phishing attacks. This makes the proposed model highly scalable and computational with results that are better suited for real time computational in cloud environment for real time analysis of phishing episodes across large datasets. These contributions enhance the model’s capability of detecting phishing URLs while at the same time offer a robust, scalable solution that can be adapted for other uses in cybersecurity domain. The research concludes that the proposed MHA-BiGRU model, optimized with DAOA, significantly outperforms traditional models in phishing detection, achieving an accuracy of 98.85%. The model’s high precision, recall, and F1-score confirm its reliability and robustness. The effectiveness of the proposed method can be empirically validated across diverse datasets. In future work, this approach could be expanded to mitigate phishing fraud on blockchain platforms. Furthermore, we aim to define generalizability metrics for the models, ensuring their adaptability to a wide range of scenarios.

Supplemental Information

Supplemental Information 1 Block diagram of cloud computing-based IOV architecture.

Supplemental Information 2 Average of the DAOA-DLPC technique on 80% TAS and 20% TSS.

Supplemental Information 3 Loss curve of the DAOA-DLPC system at 80% TRAS and 20% TESS.

Supplemental Information 4 PR curve of the DAOA-DLPC method at 80% TRAS and 20% TESS.

Supplemental Information 5 Average of the DAOA-DLPC model under 70% TRAS and 30% TESS.

Supplemental Information 6 Comparative outcomes of DAOA-DLPC system with recent systems.

Supplemental Information 7 Details of the dataset.

Additional Information and Declarations

Competing Interests

The authors have no competing interests.

Author Contributions

Vishnukumar Ravula conceived and designed the experiments, performed the experiments, performed the computation work, prepared figures and/or tables, authored or reviewed drafts of the article, and approved the final draft.

Mangayarkarasi Ramaiah conceived and designed the experiments, analyzed the data, prepared figures and/or tables, authored or reviewed drafts of the article, and approved the final draft.

Data Availability

The following information was supplied regarding data availability:

The data is available at Mendeley: Tan, Choon Lin (2018), “Phishing Dataset for Machine Learning: Feature Evaluation”, Mendeley Data, V1, doi: 10.17632/h3cgnj8hft.1.

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
