# Peer review of "Enhancing phishing detection with dynamic optimization and character-level deep learning in cloud environments"

_PeerJ Computer Science, doi:10.7717/peerj-cs.2640_

## Round 0.1 · original submission · Major Revisions

Dear authors,

The reviews for your manuscript are included at the bottom of this letter. We ask that you make necessary changes and additions to your manuscript based on those concerns and criticisms. Furthermore, adding a discussion for synthesis of findings, implications, future research, and limitations will be better.

Best wishes,


Reviewer 1 ·

Basic reporting

1.The abstract should be improved. Your point is your own work that should be further highlighted.
2.Introduction seems to be incomplete. Please carefully check and supplement it.
3. More statistical methods are recommended to analyze the experimental results.
4. The article can be further enhanced by connecting the undergoing work with some existing literatures.
5. The numerical simulation verification is not convincing, and the actual engineering application example verification should be given.
6. There are a few typos and grammar errors in the manuscript.

Experimental design

As above

Validity of the findings

As above

Additional comments

As above

Reviewer 2 ·

Basic reporting

See below

Experimental design

See below

Validity of the findings

See below

Additional comments

1. English should be improved.

2. Title: Please use the literature background to develop a better title in order to catch the prospective reader's attention.

3. Abstract: It should be rewritten in terms of aim, background, motivation, and significant results. Your abstract should clearly state the essence of the problem you are addressing, what you did, and what you found and recommended. That would help prospective readers of the abstract decide if they wish to read the entire article.

4. The main concern I have about the paper is with respect to the contributions of your work. The methodology used has few outstanding innovation points. So what innovative work have you done compared with previous studies?

5. How does your work contribute to this field? It is not clearly stated in the abstract and conclusion sections.

6. Introduction: it is not well organized. With the current introduction, it is not possible to understand the gap between the previous research and the novelty of the current works. The introduction should be fully reorganized to show the difference between the current work and the previous one.

7. What is the new devolved methodology? It is not clearly presented.

8. The provided results are limited, not convincing, and insufficient to publish in this quality journal.

9. Discussion of the results should provide useful insights. The results should be further elaborated to display how they could be utilized in real-world situations. The authors should further develop critical assessment in the results discussion.

10. Conclusions: It is poorly written and lacks an evaluation of the significant and main findings from the current work that have been found.

Reviewer 3 ·

Basic reporting

Authors introduce a Dynamic Arithmetic Optimization Algorithm with Deep Learning-Driven Phishing URL Classiûcation (DAOA-DLPC) algorithm in cloud infrastructure. The presented technique focuses on the detection and classiûcation of phishing based on URL websites.

1. In the abstract, we advised to rewrite in order to reflect the significance and provide the point and quantitative advantages of your work.
2. The manuscript's motivations should be further highlighted in the manuscript, e.g., what problems did the previous works exist? How to solve these problems? The authors may consider analyzing the problems of the previous works and how to address these problems with the proposed method. Please explain that.
3. Some future work should be clarified in Conclusion.

Experimental design

4. The results should be compared with other recent methods. The reviewer would say that there should have other efficient approaches.

Validity of the findings

5. The authors must clearly explain the difference(s) between the proposed method and similar works in the introduction.

Reviewer 4 ·

Basic reporting

1. The entire paper contains several ambiguity in writing, including the abstract. Many words are not separated by spaces, for ex - in line 20 it's written 'frameworkallows' instead of 'framework allows'.
2. The abstract needs to be re-written with clarity. It can be made a little shorter as well. Some portion of the abstract can be moved to the introduction section.
3. The introduction also looks totally unstructured. It looks like some chunks are just combined together. It should display a brief background of the problem under consideration, limitations of existing approaches, motivation for this work and main results.
4. The contribution of this paper is also not stated clearly.
5. The works described in 'Related Work' section need re-organization.
6. Figures are not clear, blurry and lacks quality.
7. The inline mathematical terms and equations are not formatted properly. They should be re-written and follow LaTeX equation style.

Experimental design

1. Proper details on the experiments should be provided along with the description and features of the dataset used in this paper.
2. The parameters used in the experiment are not well-explained.
3. The steps of the experiment are also not clearly explained and are difficult to follow.
4. The method is not assisted by a formal architecture. The design of the mechanism should be illustrated via a proper end-to-end architecture.

Validity of the findings

1. The study does not consider the comparison of the proposed approach with baselines.
2. Rather than using generic classifiers, the experiments should be performed on standard models for performance evaluation.
3. Robustness and ablation studies should be included.
4. The results lack proper justification.

---

## Round 0.2 · Minor Revisions

Dear authors,

Thank you for the revision. One of the original reviewers did not respond to the invitation for reviewing the revised paper. According to one of the final reviewers, your paper still needs a revision and we encourage you to address the concerns and criticisms of Reviewer 2 and resubmit your article once you have updated it accordingly.

Best wishes,

Reviewer 1 ·

Basic reporting

This paper can be accepted now.

Experimental design

This paper can be accepted now.

Validity of the findings

This paper can be accepted now.

Reviewer 2 ·

Basic reporting

See below

Experimental design

See below

Validity of the findings

See below

Additional comments

English should be improved.
Title: Please, use the literature background to develop a better title in order to increase catch the prospective reader’s attention.
Abstract: It should be rewritten in terms of aim, background, motivation, and significant results. Your abstract should clearly state the essence of the problem you are addressing, what you did and what you found and recommend. That would help prospective readers of the abstract to decide if they wish to read the entire article.
The main concern I have about the paper is with respect to the contributions of your work. The used methodology has few outstanding innovation points. So what innovative work have you done compared with previous studies?
How does your work contribute to this field? It is not clearly stated in the abstract and conclusions sections.
Introduction: it is not well organized. With current introduction is not possible to understand the gap of the previous research and novelty of the current works. The introduction should be fully reorganized to show the difference between the current work and previous one.
What is the new devolved methodology? It is not clearly presented.
The provided results are limited, not convincing and insufficient to publish in this quality journal.
Discussion of the results should provide useful in-sights. Where, the results should be further elaborated to display how they could be utilized in real uses. The authors should further grow critical assessment in results discussion.
Conclusions: it is poor written and lacks the evaluation of the significant and main findings from the current work that has been found.

Reviewer 3 ·

Basic reporting

THIS PAPER CAN BE ACCEPTED NOW.

Experimental design

THIS PAPER CAN BE ACCEPTED NOW.

Validity of the findings

THIS PAPER CAN BE ACCEPTED NOW.

---

## Round 0.3 · Minor Revisions

Dear Authors,

Thank you for the reivised paper. In accordance with the comments provided by the two initial reviewers, it is recommended that the paper undergo a minor revision. It is encouraged that the identified concerns and criticisms be addressed and that the article be resubmitted once the requisite updates have been made.

Best wishes,

Reviewer 2 ·

Basic reporting

See below

Experimental design

See below

Validity of the findings

See below

Additional comments

* In the abstract, adding one sentence discussing the results will help the reader to understand the significance of this work.
* The reasons and significance of this study should be clearly defined by highlighting the limitations of the previous studies in this field. Following their discussion of the shortcomings of previous studies, the authors need to explicitly list all of their contributions.
* What are the contributions of this study? The authors should mention their contributions or significance of this work in the introduction.
* Please provide a paragraph immediately before Section 2 containing the contexts of this work.
* A comparison table can be included in the results section to highlight the merits of the proposed strategy considering various factors.

Reviewer 4 ·

Basic reporting

1. Some sentences are hard to read for example - rewrite lines 41, 43, 269-271, 292, eq. 3, line 296, etc.
Also, recheck the manuscript for grammatical errors.
2. Rewrite literature review paras to include the references in the beginning of each paragraph, like done in line 116.
3. The figures are still not clear and blurry.

Experimental design

-

Validity of the findings

-

Additional comments

-

---

## Round 0.4 · Minor Revisions

Dear Authors,

Thank you for submitting your revised article. Feedback from the reviewers is now available. There are still some important issues that have not been addressed by two revisions. We strongly recommend that you clearly address the issues raised by Reviewer 4, pay special attention to them and resubmit your paper after making the necessary changes.

Best wishes,

Reviewer 2 ·

Basic reporting

accept

Experimental design

accept

Validity of the findings

accept

Additional comments

accept

Reviewer 4 ·

Basic reporting

1. The writing has been improved, but still having some errors. For example, lines 68-70 contain grammatical errors. There are still some chunks in literature review that are not clearly written. For example, line 146 contains a fullstop after the citation. The same goes for line 152,162,168 and so on. Therefore, this section still needs refinement.
2. The authors claim in their rebuttal that the image quality has been improved. But there are no visible changes in the quality of the images especially in Fig 4. The number of epochs are not visible and is still faded.
3. The Table 6 is actually Table 5? Recheck the table numbers. Use LaTeX to draw Table 5/6 (on last page).

Experimental design

-

Validity of the findings

-

Additional comments

-

---

## Round 0.5 · accepted · Accept

Dear Authors

Thank you for addressing the reviewers' comments. The manuscript now seems ready for publication.

Warm regards,

Reviewer 4 ·

Basic reporting

-

Experimental design

-

Validity of the findings

-

Additional comments

-